# Professionals’ Perspectives on the Effects of the COVID-19 Pandemic among Child and Adolescent Victims of Domestic Violence Living in the Portuguese Residential Foster Care System

**DOI:** 10.3390/ijerph20105826

**Published:** 2023-05-15

**Authors:** Micaela Salgado, Sofia Neves, Estefânia Silva

**Affiliations:** 1Department of Social and Behavioral Sciences, University of Maia, 4475-690 Maia, Portugal; a039240@umaia.pt (M.S.); egsilva@umaia.pt (E.S.); 2Interdisciplinary Center for Gender Studies (ISCSP-ULisbon), 4475-690 Maia, Portugal

**Keywords:** COVID-19, children, adolescents, residential foster care system, professionals, domestic violence

## Abstract

The COVID-19 pandemic had several negative impacts on child and adolescent victims of domestic violence, especially on those who lived in the residential foster care system. The main goal of the present study was to understand these negative impacts through the perspectives of professionals in Portuguese residential foster care structures using both individual interviews and an online survey. One hundred and three professionals aged between 22 and 64 years (M = 38.39; SD = 8.34) participated in the online survey (86 females and 17 males). Of those, seven professionals, four females and three males aged between 29 and 49 years (M = 38.43, SD = 7.50), were also interviewed. According to the participants, the conditions imposed by the COVID-19 pandemic contributed adversely not only to the increase in domestic violence against children and adolescents but also to the aggravation of the conditions children and adolescents living in the Portuguese residential foster care system were exposed to, namely concerning family relationships, access to resources and services, and institutional dynamics. The results suggest the necessity to develop standard procedures to cope with pandemic situations in the residential foster care system.

## 1. Introduction

SARS-CoV-2, which causes Coronavirus Disease (COVID-19), was identified in Portugal in March 2020. In the same month, the government declared the first State of Emergency [1], which produced several changes in the daily life of persons and institutions. Measures requiring physical distance, safety rules concerning circulation and lockdowns, and the closure of schools and public and private services and businesses [2] resulted in individual and collective challenges, with significant impacts on health and well-being.

International research [3,4] has concluded that pandemic phenomena, humanitarian crises, and natural disasters, due to containment policies, may increase the vulnerabilities of victims of violence, specifically those who are victims of domestic violence (DV). A systematic review of DV during the COVID-19 pandemic [5] revealed that home lockdown left the victims more exposed to their perpetrators, occasioning increased violence and diminished reports. A study conducted in Portugal [6] showed that in a sample of 1062 participants, 13.7% (*n* = 146) stated that they had suffered DV during the pandemic. Most victims were subjected to psychological violence (13.0%, *n* = 138), with 1.0% (*n* = 11) and 0.9% (*n* = 10), respectively, exposed to sexual and physical violence. Lower age was identified as a risk factor, although both sexes and different age groups experienced DV. 

Some studies have pointed out an association between the COVID-19 pandemic and the expansion of the reported cases of child abuse [7]. The evidence seems particularly prominent in DV situations, which can be explained by pandemic-related stressors, specifically in the context of family, and difficulty in benefiting from protective resources [8]. 

A review of early studies concerning COVID-19 and violence against children [9] highlighted four key conclusions: (1) a decrease in police reports and referrals to child protective services was observed, (2) it was not possible to affirm that the number of calls to police or domestic violence helplines increased or decreased, (3) more child abuse-related injuries were treated in hospitals compared with number of injuries pre-COVID-19, and (4) an increase in family violence was recorded.

Although the placement of children into foster homes declined by half during the pandemic [7], it has been demonstrated that this did not correspond to a reduction in DV perpetrated toward children nor a diminution of the responsibilities of those involved in the child welfare system [10,11]. By contrast, child protective and residential foster care services felt the impact of the restrictions on their functioning, with professionals expressing that they were subjected to additional duties and further stress [12]. Secondary traumatic stress, burnout, and compassion satisfaction in foster care workers during the COVID-19 lockdown were identified [13]. 

In the same way, children and adolescents living in foster institutions saw their routines drastically change, principally during periods of quarantine, when schools were closed. The limited access to health facilities as well as other services (e.g., evidence-based services and crisis services), along with the obligation to supervise children and adolescents in all their daily tasks, imposed additional challenges on professionals, children, and adolescents [14,15]. Pandemic stressors exacerbated most existing problems and created new ones [16,17]. The adverse consequences of the COVID-19 pandemic were felt, for example, by current and former foster youth aged between 18 and 23 years old who reported clinical levels of mental health issues, food insecurity, unemployment, and becoming homeless [18]. In addition, foster youth aged 18 to 26 described intensified levels of concern about finances, health, social support, well-being, and professional achievements [19]. A survey conducted on a sample of 2117 young people showed decreased employment rates and increased disconnection between school and work [20]. Additionally, young parents with foster care backgrounds revealed abrupt loss of employment, educational challenges, insufficient resources, barriers to service receipt, and parental and child mental health worries [19].

### Residential Foster Care System and DV in Portugal during the COVID-19 Pandemic

At the beginning of the COVID-19 pandemic, the General Directorate of Health, the entity responsible for defining the measures aiming to prevent and control the disease in Portugal, determined that the admission of children and adolescents to the foster care system should respect the mandatory procedures regarding testing for SARS-CoV-2 and, in positive cases, the 14-day prophylactic isolation (Orientation 009/2020, 11 March). These procedures would be revoked later, recognizing that the 14 days of prophylactic isolation and the prohibition of contacting families could harm children’s and adolescents’ well-being.

Scientific research on DV against children and adolescents during the COVID-19 pandemic and its impact on the Portuguese foster care system is scarce. Besides the evidence that the number of official complaints concerning DV crimes decreased in 2020 (27,637) and 2021 (26,520) compared with 2019 (29,498), the information regarding DV against children and adolescents is limited [21]. Nearly 16% of the victims of DV are under 16 years old, and 20% are offenders’ descendants or stepchildren. Concerning child sexual abuse, most victims are aged between 8 and 13 years old. Approximately 50% of the abusers are members of the victims’ families. 

The Portuguese Law on the Protection of Children and Young People in Danger (Law 142/2015 of 8 September) favors the placement of the child in his/her familiar environment [22,23], whether it is the natural or foster care family. 

In 2020, the National Commission for the Promotion of the Rights and the Protection of Children and Young People (CNPDPCJ), the national public entity in charge of the coordination of all actions regarding the promotion of the rights and the protection of children and young people in Portugal, followed 66,529 cases, of which 13,363 (32.3%) were related to DV situations [24] and 11,955 (28.9%) were neglect situations. Only 4.7% of the measures applied were in the residential foster care system, compared with 6.6% in the previous year. In 2021, 69,727 cases were followed by the CNPDPCJ, focusing on, once again, DV situations [25]. In 2021, of the total of children and adolescents living in foster care, 96.5% were in residential foster care and 3.5% were in family foster care [26]. Most children and adolescents were male (52%) and aged between 15 and 17 years old (33.5%). 

Considering a period of 10 years, 2020 was characterized as the year in which the lowest number of children entered the national residential foster care system [27]. A 4.7% and a 4.5% decrease were observed in 2020 and 2021, respectively, compared with 2019 (*n* = 896). The fall in the rates may be related to the COVID-19 pandemic and, mainly, the mandatory lockdown that occurred in 2020, a social condition that made it difficult to identify situations of violence [18].

To analyze adolescents’ psychological adjustment across two assessment waves, before and during the pandemic, Costa et al. [28] evaluated 243 adolescents living in 21 Infant and Juvenile Residential care institutions. The authors concluded that cohesion predicted negative emotional distress, confirming the importance of residential care members supporting each other and creating positive contextual interactions [28].

Justified by the absence of studies characterizing the negative impacts on child and adolescent victims of domestic violence living in the Portuguese residential foster care system during the COVID-19 pandemic, the specific aims of this study were to (a) characterize the prevalence and dynamics of DV; (b) analyze the consequences of the COVID-19 pandemic on the functioning of the national residential foster care system; and (c) understand the impacts of the restrictions imposed by the COVID-19 pandemic on the lives of child and adolescent victims of DV in the foster care system. The focus was always on the professionals’ perceptions, aiming to highlight their experiences. 

## 2. Materials and Methods

### 2.1. Participants

The total sample was comprised of 103 participants (Cf. Table 1)—86 females (83.5%) and 17 (16.5%) males aged between 22 and 64 years old (M = 38.39; SD = 8.34). Most of the sample had graduated (*n* = 70; 68%), mainly in social service (*n* = 41; 39.8%), and 42.7% (*n* = 44) had worked with children and adolescents at risk for up to 10 years. The large majority carried out their functions in institutions located in the north of the country.

### 2.2. Instruments

The online survey, built from scratch on the basis of the consultation of national and international specialized literature, was composed of 33 mixed questions organized into five sections: (1) informed consent; (2) sociodemographic characterization (e.g., age, sex/gender identity, district of residence); (3) professional practice characterization (e.g., position in the institution, years of work); (4) characterization of the Portuguese foster care system during the COVID-19 pandemic (e.g., main reasons for domestic violence before and after the pandemic, main aggressor(s), signs and symptoms most often presented by victims before and after the pandemic); and (5) observations. The online survey was first tested by two professionals to guarantee that the questions were comprehensible. 

The interview script constituted 28 mixed questions based on the literature and was divided into five sections: (1) informed consent; (2) sociodemographic characterization (e.g., age, education, professional experience); (3) professional practice characterization (e.g., how do you characterize the residential foster care where you work?); (4) characterization of the residential foster care system and the foster population during the COVID-19 pandemic (e.g., what transformations did you observe in institutions with the emergence of the pandemic?); and (5) characterization of the impact of the COVID-19 in DV cases (e.g., what do you think that was the impact of the pandemic on the phenomena of DVCA?; what might have been the impact of mandatory confinement in family contexts where DV already existed?). 

### 2.3. Procedure

The procedure complied with the Code of Ethics and Deontology of the Portuguese Psychologists Association (Código de Ética e Deontologia da Ordem dos Psicólogos Portugueses) and the American Psychology Association (APA). The General Regulation on Data Protection (Regulamento Geral de Proteção de Dados) of the European Union was followed.

To obtain as many answers as possible, an email was sent to all 310 Local Commissions for the Protection of Children and Young People explaining the purpose of the study and inviting eligible participants to collaborate. The study and the link to the online survey were also published and disseminated on social networks (e.g., Facebook and LinkedIn).

To be eligible to participate, individuals needed to have experience working with children and adolescent victims of DV living in the Portuguese residential foster system during the critical period of the COVID-19 pandemic, i.e., the year 2020. All potential participants were asked to fill out the online survey. Those who intended to be interviewed also needed to send an email to the research team expressing their availability. 

The interviews were conducted via digital platforms, mostly Zoom, in accordance with the participants’ interests. All participants validated the informed consent and were authorized to be audio-recorded. Data were collected between November 2021 and March 2022.

### 2.4. Data Processing and Analysis Techniques

Descriptive statistical analysis was performed using IBM SPSS (New York, NY, USA) software version 28 to describe the basic features of the quantitative data in the study. 

Qualitative data were subjected to a thematic analysis, following the six-phase methodology authored by Braun and Clarke (2006). After the full transcription, the entire content of the interviews was considered in the process of categorization, which was first conducted by three researchers separately. Afterward, the primary categories and codes were presented and discussed in two analysis meetings involving the whole research team. Finally, a consensus approach was reached, resulting in a final and common codification. 

The principal results of the study are described in the following section. 

## 3. Results

### 3.1. Quantitative Data

Overall, participants affirmed that there was no increase in the number of children and adolescents in the Portuguese residential foster care system during 2020 (*n* = 81; 78.6%). Considering the children and adolescents who were fostered during the critical period of the COVID-19 pandemic, 92.2% (*n* = 95) of participants revealed having noticed signs of sadness, 91.3% (*n* = 94) noticed signs of anxiety, 87.4% (*n* = 90) noticed opposition/challenging behaviors, 86.4% (*n* = 89) noticed signs of irritability, 84.5% (*n* = 87) noticed learning difficulties, and 73.8% (*n* = 76) noticed sleep difficulties. 

According to participants, children and adolescents were mainly exposed to neglect (*n* = 84; 81.6%), psychological violence (*n* = 39; 37.9%), and physical violence (*n* = 23; 22.3%). DV was mostly perpetrated by fathers (*n* = 82; 79.6%) and mothers (*n* = 68; 66%).

### 3.2. Qualitative Data

From the total sample, only seven participants were accepted for interviews. The thematic analysis resulted in four central categories (Cf. Figure 1): (1) the impacts of the COVID-19 pandemic on DV cases; (2) the impacts of the COVID-19 pandemic on the Portuguese residential foster care system; (3) the impacts of the COVID-19 pandemic on child and adolescent victims of domestic violence living in the Portuguese residential foster care system; and (4) the resources and responses of the professionals working in the Portuguese residential foster care system during the COVID-19 pandemic. To illustrate the categories, excerpts of the interviews are presented.

#### 3.2.1. The Impacts of the COVID-19 Pandemic on DV Cases 

According to participants, the COVID-19 pandemic led to an amplification of DV cases in terms of its reiteration and severity. Both intimate partner violence and violence against children and adolescents were boosted by the amount of time people were forced to spend together under unknown circumstances. This was particularly true in periods of lockdown due to its stressful nature. Families had to face demanding challenges in both personal and professional domains, with additional tasks requiring more attention and investment. In many situations, labor conditions worsened or became more intense, with adults performing remote work at home and simultaneously assisting children and adolescents in school activities. In addition, as unemployed increased, economic difficulties deteriorated, putting adults’ emotional stability at risk.


*“I think lockdown (…) has brought more domestic violence into homes (…). I believe that most of the cases were hidden (…). I would say that it was an atrocious increase. (…) “It brings a lot of frustration and I think it increased the level of domestic violence at home between parents and children because tolerance was much lower (…)”*
(I2-M-40)


*“(…) but then there it is, the continuity, all those hours, 24 under 24 together ended up triggering even more.”*
(I1-M-49)

Additionally, because entities were closed or limited to minimal services, DV cases became difficult to detect. 


*“The fact that we were all confined, there were no field teams that could prevent issues of domestic violence”*
(E2-M-40)


*“(…) the children, if they are not at home, where are they? They’re in school! Therefore, if the school is closed and the children must be automatically at home, no one does this, this lookout and this, that take care of the child.”*
(I7-F-29)

#### 3.2.2. The Impacts of the COVID-19 Pandemic on the Portuguese Residential Foster Care System

The consensus among participants was that the COVID-19 pandemic had severe effects on the Portuguese residential foster care system. Drastic changes were observed not only concerning internal institutional dynamics but also interinstitutional articulation. The adoption of safety and protective measures to control the disease and prevent its propagation, such as social distancing, respiratory etiquette, testing, prophylactic isolation, and mandatory quarantine, caused disturbances in the teams and the children and adolescents, provoking a generalized sense of fear and dissatisfaction. The closure of schools was perceived as particularly demanding to professionals who had to support children and adolescents’ daily activities. Furthermore, the restrictions regarding family contact were identified as one of the factors causing more disruption in the relationship between the teams and the children and adolescents. 


*“(…) we are aware that these types of restrictions are not advantageous for children, because it limits the intervention a little bit and limits the interaction between family members and children.”*
(I7-F-29)

Concerning interinstitutional articulation, participants pointed out the constraints felt in terms of the Promotion and Protection Processes. Because of the successive State of Emergency declared by the President of the Portuguese Republic, which imposed lockdown measures, it is likely that fewer risk and danger cases were detected. 

#### 3.2.3. The Impacts of the COVID-19 Pandemic on Child and Adolescent Victims of Domestic Violence Living in the Portuguese Residential Foster Care System

Children and adolescents living in the Portuguese residential foster care system were, from the perspective of the professionals, the most affected by the measures imposed during the lockdown periods. Once again, questions regarding the closure of the schools and the obligation to take part in a distance learning approach were expressed as especially tense and problematic, aggravating anxiety problems and troubling conduct. Being in a social distancing regime was particularly difficult, as it limited the interaction with peers and families, putting children and adolescents’ mental health at risk. The period of the 14-day prophylactic isolation was indicated as the most critical.


*“(…) They are tired of being closed and together (…) it´s the same to us, adults, isn’t it? If we are too much time with the same people, we get tired, isn’t it? So, sometimes we need to be away for a while and the children need that too, isn’t it? They were 20 children living here daily (…).”*
(I6-F-46)

#### 3.2.4. The Resources and Responses of the Professionals Working in the Portuguese Residential Foster Care System during the COVID-19 Pandemic

Given the aforementioned impacts of the lockdown and the greater vulnerability of children and adolescents living in the foster care system during that period, the participants shared three main strategies that they used to help them cope with social distancing and isolation: (1) encouraging alternative communication through electronic and digital means; (2) improving affective relationships between professionals and children and adolescents through trust and confidence; and (3) promoting entertainment and distraction by organizing ludic activities. 


*“Here we always encourage contact to be maintained. (…) Always, the telephone contact, and the video calls, we promote here that contact should remain. (…) And to the youths themselves keep them abreast of the process and what’s happening, I think that’s fundamental”*
(I4-F-39)


*“And then also, greatly increase the number of (…) playful-pedagogical activities (…) inside the house to keep them also active and distracted as well and fun (…)”*
(I3-F-36)

## 4. Discussion

The findings of the present study emphasize the negative impacts of the COVID-19 pandemic on children and adolescents living in the Portuguese residential foster care system, especially during lockdown periods. The most preeminent consequences, besides the changes in internal and external institutional processes and dynamics, were felt by children and adolescents who were deprived of school routines, regular social interactions, and family contact and those who were subjected to measures of prophylactic isolation considered particularly stressful. As indicated by several studies, e.g., [28], they experienced manifestations of sadness, signs of anxiety, opposition/challenging behaviors, signs of irritability, learning difficulties, and sleep difficulties, showing that children and adolescents’ mental health was highly disturbed.

Corroborating a study by Machlin et al. [29], the present study revealed that the pandemic situation, mainly due to mandatory lockdowns, caused a perceived increase in the phenomena of DV, explained, among other factors, by intra-family challenges and difficulties in accessing support structures and institutional services [8]. Although the extent and long-term impacts of this crisis remain largely unknown, several transformations were observed in the organization of social life. The obligation to follow a rigorous protocol to prevent and combat a disease that caused the deaths of thousands of people triggered generalized feelings of tension, hopelessness, and fear. In this context, it is also relevant to mention that Portugal is considered the country with the most cases of and deaths from COVID-19 (per million inhabitants) in the European Union and the second most cases and deaths in the world [30]. Children and adolescents were particularly disturbed by these feelings, as were their caregivers, who simultaneously felt overwhelmed with the overload of tasks and responsibilities [31,32]. 

In addition, the instability generated by lockdowns was motivated by economic constraints, unemployment or precarious job situations, and the loss of financial autonomy. As schools were closed, family challenges became more demanding, enhancing parental frustration, impatience, burnout, and, consequently, situations of tension and violence [33,34,35]. Thus, children and adolescents became more vulnerable to DV since parental burnout is associated with an increased risk of violence and neglect [36].

Along with the participants’ perceptions of DV exacerbation, our findings also demonstrate that the lockdown periods significantly amplified the condition of secrecy often linked to this crime. The need for home lockdowns, with the prohibition of social contact, prompted an increase in the amount of time and proximity between aggressors and victims. This condition, in turn, intensified victims’ vulnerability and reduced their ability to report [37]. 

On the other hand, the mandatory remote work imposed by lockdowns and the low availability of the first-line entities providing support for victims reduced the capability to signal DV cases [38]. A report presented by the CNPDPCJ [24] highlighted the need to carefully analyze the rates of children and adolescents referred to the Portuguese foster care system during the critical phase of the pandemic, as they could not represent the reality of the violence perpetrated against them. 

In addition, during the State of Emergency, fewer police authority interventions took place, and some jurisdiction and child protection services were forced to reduce their operation or close [39], which made it difficult to report or identify these crimes and facilitate their recidivism [40]. This might be one of the reasons why an increase in residential foster situations was not observed throughout mandatory lockdowns. According to our participants, the entire system was affected by the measures implemented by the government, although they recognized their necessity. Thus, the foster care system was no exception. In fact, despite the negative consequences of the lockdown measures, the early action of the Portuguese government, compared with those of other countries, reduced overall mobility by 80% [41].

As documented in the CASA Report, 2020 was the year in which the lowest number of children and adolescents entered the foster care system [27]. The diagnosis of situations of danger during the State of Emergency was one of the greatest difficulties felt by institutions [24]. Consequently, the application and promotion of protective measures were limited [42,43,44].

According to evidence from previous years, DV is characterized as neglect and psychological abuse in most cases [27]. Perpetrated commonly by parents, the male figure stands out as the most expressive, suggesting that families are still trapped in a patriarchal regime in which children and women are the most defenseless [42].

Due to these adversities, aligned with the needs of foster children and adolescents [45], the work carried out by professionals implied additional efforts. Besides regular activities, strategies to promote safety and confidence and assure contact with peers and families were carried out. As Bernardi [46] pointed out, the adoption of these strategies might have had a positive effect on the mitigation of the traumatic nature of the lockdown. In the same way, from the perspectives of professionals, it was also crucial to promote safe relationships with children and adolescents to create a better residential environment and improve moments of individual attention and open communication, allowing the understanding of potential concerns and the clarification of some eventual misinformation [47]. 

These resources and responses from professionals were recommended by the Portuguese Psychologists Order [48] and validated as beneficial by Montserrat et al. [49], who stated their relevance to the well-being of children and adolescents living in the residential foster care system.

## 5. Conclusions

The present study aimed to contribute to a more comprehensive approach to understanding the effects of the COVID-19 pandemic among child and adolescent victims of domestic violence living in the Portuguese residential foster care system through the perspectives of professionals who directly worked with them. The results suggest that the perceived consequences were characterized as pervasive and devasting in terms of mental health and well-being, not only for children and adolescents but also for themselves. The closure of schools and the prohibition to visit families were the most damaging effects felt by children and adolescents, and the overload of tasks was the most damaging effect for professionals. 

Although the number of children and adolescents entering the system did not increase in the most critical phases of the pandemic, there is a global perception that this was not due to a reduction in DV cases. The low efficacy of institutions to detect risk and danger, along with the victims’ inability to access protection mechanisms and support structures, might explain the alleged diminution of DV and, in consequence, entries into the residential foster care system. As it is not possible to affirm that an increase in the official rates of crime corresponds necessarily to a real increase in crime, it is also not possible to affirm the opposite. 

The findings indicate the need to reflect on defining contingency plans to support residential foster care services in crisis situations, with protocols and strategies to mitigate the impact of potential trauma in children, adolescents, and professionals. We believe that this study can be an important tool to promote reflection among policymakers, practitioners, and researchers on the factors that need to be considered in future pandemic scenarios. The constitution of a task force composed of both representatives of different areas and foster children and adolescents could be an innovative way of identifying priorities.

Some limitations were observed in this study. First, the quantitative data are not representative of all 310 Local Commissions for the Protection of Children and Young People existing in Portugal. Second, the number of individual interviews was lower than expected due to the difficulty in obtaining the agreement of professionals to participate. Justifications concerning lack of time and excess work were most common for refusal. 

Further studies are needed to better understand medium- and long-term signs and symptoms exhibited by children and adolescents must be developed as well as to comprehend how professionals working in the residential foster care system cope with the stress associated with job duties during lockdown periods. 

## Figures and Tables

**Figure 1 ijerph-20-05826-f001:**
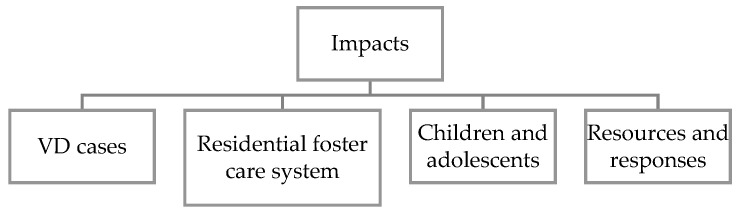
Main central categories.

**Table 1 ijerph-20-05826-t001:** Participants’ characteristics.

	*n*	%	*M*
Age			38.39
Gender			
Male	17	16.5%	
Female	86	83.5%	
Nationality			
Portuguese	103	100%	
Education			
Secondary school	6	5.8%	
Bachelor’s degree	70	68%	
Master’s degree	25	24.3%	
Ph.D.	1	1%	
Area			
Social Service	41	39.8%	
Psychology	33	32%	
Others	34	33%	

## Data Availability

The data are not publicly available due to privacy.

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
