# Peer review of "Professionals’ Perspectives on the Effects of the COVID-19 Pandemic among Child and Adolescent Victims of Domestic Violence Living in the Portuguese Residential Foster Care System"

_ijerph, 2023, doi:10.3390/ijerph20105826_

Round 1

Reviewer 1 Report

Thank you for this research. I appreciate the mixed methods approach, despite your limited qualitative data. Your findings confirmed what researchers around the globe feared during the pandemic, but it is important to highlight that it happened to Portuguese children as well. The manuscript reads fairly straightforward, but I would have liked to see additional interview responses included as well as some of the wording of your survey instrument as it was developed from "scratch". I would have also liked to see if there was a cultural explanation for the rise of DV beyond the obvious explanation of we were in a global pandemic and stressors were heightened and resources were tapped out. In other words, was there anything unique happening in this country that was not happening world wide to cause a rise in DV?  Also, are there any plans of follow up to see if DV rates have declined post pandemic?

Author Response

Dear reviewer

Thank you so much for all your comments and suggestions.

Please see our answers below.

Best

The authors

_________

Thank you for this research. I appreciate the mixed methods approach, despite your limited qualitative data. Your findings confirmed what researchers around the globe feared during the pandemic, but it is important to highlight that it happened to Portuguese children as well.

The manuscript reads straightforward, but I would have liked to see additional interview responses included as well as some of the wording of your survey instrument as it was developed from "scratch".

Answer: Thank you very much for your comment. We integrated additional information about the interview responses and the survey instrument.

I would have also liked to see if there was a cultural explanation for the rise of DV beyond the obvious explanation of we were in a global pandemic and stressors were heightened and resources were tapped out. In other words, was there anything unique happening in this country that was not happening worldwide to cause a rise in DV? 

Answer: Data from all around the world shows an increasing of domestic violence during the critical period of the pandemic, despite the reduction of official complaints.

Also, are there any plans of follow up to see if DV rates have declined post pandemic?

Answer: The Portuguese Government collect data on criminal records and publish annual information on the characteristics of domestic violence. The information includes the comparison of years, aiming to analyze the trend of the phenomena.

Reviewer 2 Report

The paper represents a very important scientific contribution on the impact of specific circumstances during the COVID pandemic and measures used to prevent the spread of the disease. I believe that paper like this will help in interpreting the long-term consequences of the COVID pandemic, which we can still expect.

Abstract: emphasize that 103 participants are participants in the qualitative study.

Materials and Methods: This entire passage lacks a clear distinction between quantitative and qualitative research. Please show this part systematically. Also, since the quantitative analysis preceded the qualitative one, they should be mentioned in that order.In this whole paragraph, you can apply some of the published guidelines for displaying qualitative data.

2.1 Participants: a description of the quantitative research of the participants is missing

2.2 Instruments: replace the order of presentation of instruments for quantitative and qualitative research

Qualitative research: on which basis were the questions formed (on the basis of earlier research, on the basis of the literature, on the basis of the quantitative research of this study)?

Quantitative research: was the questionnaire piloted before conducting the online survey?For each question section, examples of questions are missing or at least descriptive ones, eg sociodemographic characterization: age, gender, etc. Are the questions open or closed? If any answers were offered, what were they?

2.3 Procedure

Do you have information on how many people are employed in the Local Commissions for the Protection of Children and Young People whom you sent an invitation to participate?Did you send e-mails to employees individually or did you send e-mails to one official address? Have you contacted the Local Commissions for the Protection of Children and Young People to establish if information about the research has reached the staff? If so, please specify. If you haven't, it can be a suggestion for future research, in order to gather as many participants possible.

Here I conclude that online research (quantitative) was conducted first, and then the interviews were organized. If I concluded wrongly, please indicate the procedure more clearly. Also, first mention the online research, which is offered to everyone, and then mention inclusion in interviews.

What exactly does dissemination of research through social networks mean? Is it just a notification about the implementation or is the link to the online survey also published on social networks? Which social networks published the content? (line 160-161)

What does "safety reasons" mean when conducting an interview (line 162)? Were the cameras off? Recorded conversations during interviews or focus groups are available only to researchers, and during the transcript, no personal data is provided, so that the identity of the participants is protected and, in a way, the participation is anonymous. If you did not apply any other participant protection measures, then this should not be particularly emphasized.

Data processing and analysis techniques: align the order with the order of research.

Author Response

Dear reviewer

Thank you so much for all your comments and suggestions.

Please see our answers below.

Best

The authors

__________

The paper represents a very important scientific contribution on the impact of specific circumstances during the COVID pandemic and measures used to prevent the spread of the disease. I believe that paper like this will help in interpreting the long-term consequences of the COVID pandemic, which we can still expect.

Abstract: emphasize that 103 participants are participants in the qualitative study.

Answer: Thank you so much for your comments. We included this information in the abstract.

Materials and Methods: This entire passage lacks a clear distinction between quantitative and qualitative research. Please show this part systematically. Also, since the quantitative analysis preceded the qualitative one, they should be mentioned in that order. In this whole paragraph, you can apply some of the published guidelines for displaying qualitative data.

Answer: We made the changes suggested.

2.1 Participants: a description of the quantitative research of the participants is missing

Answer: We had clarified this point in the document.

2.2 Instruments: replace the order of presentation of instruments for quantitative and qualitative research

Answer: As the seven participants who were interviewed also had participated in the survey, quantitative data come first.

Qualitative research: on which basis were the questions formed (on the basis of earlier research, on the basis of the literature, on the basis of the quantitative research of this study)?

Answer: We had clarified this point in the document.

Quantitative research: was the questionnaire piloted before conducting the online survey

For each question section, examples of questions are missing or at least descriptive ones, e.g., sociodemographic characterization: age, gender, etc.

Answer: The survey was firstly tested with two professionals.

We added some examples of questions to clarify the section.

Are the questions open or closed?

If any answers were offered, what were they?

Answer: We included information about sociodemographic characterization. We also had clarified the type of questions.

2.3 Procedure

Do you have information on how many people are employed in the Local Commissions for the Protection of Children and Young People whom you sent an invitation to participate?

Did you send e-mails to employees individually or did you send e-mails to one official address?

Have you contacted the Local Commissions for the Protection of Children and Young People to establish if information about the research has reached the staff? If so, please specify. If you haven't, it can be a suggestion for future research, in order to gather as many participants possible.

Answer: We don´t have information on the number of professionals employed in Local Commissions for the Protection of Children and Young People.

The emails were sent to official addresses. Further contacts were not established. Thank you for your suggestion.

Here I conclude that online research (quantitative) was conducted first, and then the interviews were organized. If I concluded wrongly, please indicate the procedure more clearly. Also, first mention the online research, which is offered to everyone, and then mention inclusion in interviews.

Answer: Your conclusion is correct. We clarified this point in the document.

All potential participants were requested to fill in the link to the online survey.

What exactly does dissemination of research through social networks mean? Is it just a notification about the implementation or is the link to the online survey also published on social networks? Which social networks published the content? (line 160-161)

Answer: We better explained the dissemination of research via social networks.

What does "safety reasons" mean when conducting an interview (line 162)? Were the cameras off? Recorded conversations during interviews or focus groups are available only to researchers, and during the transcript, no personal data is provided, so that the identity of the participants is protected and, in a way, the participation is anonymous. If you did not apply any other participant protection measures, then this should not be particularly emphasized.

Answer: Interviews were only audio-recorded. We made corrections in the paper.

Data processing and analysis techniques: align the order with the order of research.

Answer: We did it.

Reviewer 3 Report

The present study sheds light on the negative impacts of the Covid-19 pandemic on children and adolescents living in the Portuguese residential foster care system, especially during lockdown periods. The authors have extensively discussed the consequences of lockdowns, deprivation of school routines, social interactions, and family contacts, and prophylactic isolation on children's mental health, such as manifestations of sadness, anxiety, challenging behaviors, irritability, learning difficulties, and sleep-level difficulties. The authors have also highlighted that the pandemic situation, mainly due to mandatory lockdowns, caused a perceived increase in the phenomena of domestic violence (DV), which was exacerbated by intra-family challenges and difficulties in accessing support structures and institutional services.

The authors have presented a comprehensive review of the literature and used appropriate references to support their arguments.

The authors could have included some suggestions or recommendations for policymakers, practitioners, and researchers based on their findings. Such recommendations would have provided practical implications for the readers to address the issues highlighted in the study.

Overall, the study is well-written, and the findings are significant, especially in the context of the ongoing Covid-19 pandemic. However, the study's limitations, such as the lack of details on the data collection methods and the absence of recommendations for policymakers, practitioners, and researchers, need to be addressed in future studies. The voice of children is absent in the research.

Author Response

Dear reviewer

Thank you so much for all your comments and suggestions.

Please see our answers below.

Best

The authors

_____________

The present study sheds light on the negative impacts of the Covid-19 pandemic on children and adolescents living in the Portuguese residential foster care system, especially during lockdown periods. The authors have extensively discussed the consequences of lockdowns, deprivation of school routines, social interactions, and family contacts, and prophylactic isolation on children's mental health, such as manifestations of sadness, anxiety, challenging behaviors, irritability, learning difficulties, and sleep-level difficulties. The authors have also highlighted that the pandemic situation, mainly due to mandatory lockdowns, caused a perceived increase in the phenomena of domestic violence (DV), which was exacerbated by intra-family challenges and difficulties in accessing support structures and institutional services.T he authors have presented a comprehensive review of the literature and used appropriate references to support their arguments.

The authors could have included some suggestions or recommendations for policymakers, practitioners, and researchers based on their findings. Such recommendations would have provided practical implications for the readers to address the issues highlighted in the study.

Overall, the study is well-written, and the findings are significant, especially in the context of the ongoing Covid-19 pandemic. However, the study's limitations, such as the lack of details on the data collection methods and the absence of recommendations for policymakers, practitioners, and researchers, need to be addressed in future studies. The voice of children is absent in the research.

Answer: Thank you for your comments. The voice of children was not heard because it was not the goal of the study. We tried to make clear the study characteristics, including in the section concerning data collection.

We also added some recommendations.

Round 2

Reviewer 1 Report

Thank you for addressing my earlier comments. I believe this iteration answers my questions and integrates additional important contextual information.